# Towards Robust Neural Networks via Variance Minimizer Loss

## Abstract

Deep learning models are often evaluated under the assumption that setting random seeds ensures reproducibility and fairness. While repeating the same seed yields identical results, this form of reproducibility does not capture the variability that arises when different seeds are used. Such seed-dependent variation undermines the robustness and trustworthiness of reported results. We introduce Variance Minimizer Loss (VML), an adaptive, volatility-aware penalty that reduces stochastic fluctuation within a single training run. VML is architecture-agnostic and integrates as a drop-in replacement for the standard objective. On CIFAR-10/100 across four architectures, VML reduces across-seed accuracy standard deviation by 33–75% while keeping mean accuracy essentially unchanged. Crucially, VML achieves these gains without extra computational cost.

## 1 Introduction

Deep Learning (DL) models have become foundational across a wide range of applications, including healthcare diagnostics, autonomous systems, and financial forecasting, due to their remarkable ability to learn complex representations from large-scale data. Despite this success, achieving consistent and trustworthy performance from these models remains a significant and under-addressed challenge. Even when training conditions such as architecture, hyperparameters, and datasets are fixed, models often yield substantially different results across runs. This variability arises from algorithmic sources of randomness such as weight initialization, data shuffling, and optimizer behavior, which affect the trajectory of model training and lead to inconsistent outcomes. Recent studies have highlighted the sensitivity of deep neural networks to such stochastic factors, revealing that even minor changes in initialization can cause large deviations in final model performance (Summers & Dinneen, 2021). A common practice for controlling stochastic effects in deep learning is to fix the random seed during training. This does not directly reduce the influence of stochastic factors; instead, it determines the sequence of all random operations through pseudo-random number generators (PRNGs), ensuring that the same sequence is reproduced in every run with that seed. As a result, repeating an experiment with the same seed yields identical outcomes. However, when a different seed is used, the sequence of random operations changes, which in turn alters the training trajectory and can lead to substantially different results. Consequently, model performance remains sensitive to the choice of seed, and variability introduced by stochastic factors persists. While seed fixing enables a narrow form of reproducibility for a specific experimental setup, it does not ensure robustness in the broader sense needed for reliable conclusions across different runs (Pham et al., 2020). Secondly, another common approach is to report averaged metrics over multiple runs with different seeds. This provides a more comprehensive view of model performance by sampling across multiple PRNG sequences, thereby capturing a broader range of variability. While statistically more sound than relying on a single seed, this approach comes at a substantial computational cost. In many cases, it requires 25 or more complete training runs to obtain stable estimates (Renard et al., 2020; Bouthillier et al., 2019), making it impractical for large-scale experiments, resource-constrained environments, or real-time development settings.

Our research addresses these limitations by proposing the Variance Minimizer Loss (VML), an architecture-agnostic objective composed of base loss and Stable Loss (SL). The SL coefficient is scaled by current-to-reference volatility, smoothing the optimization trajectory without modifying the model or schedule. We evaluate VML under fixed training protocols while explicitly controlling individual sources of stochasticity (initialization, augmentation, and data shuffling) to isolate their

effects. Our analysis focused on the sensitivity of VML to two key hyperparameters: the penalty weight, which determines the strength of the variance reduction, and the application schedule, which controls when VML is introduced during training. Results show that applying VML early and maintaining it throughout training yields the best balance between variance reduction and learning efficiency, while keeping computational overhead minimal.

We validate our approach through extensive experiments on image classification tasks using CIFAR-10 and CIFAR-100 (Krizhevsky et al., 2009) with architectures including ResNet (He et al., 2016), VGG (Simonyan & Zisserman, 2015) ,MobileNet (Howard et al., 2017) and ShuffleNet (Ma et al., 2018). Our key contributions are:

1. **Variance Minimizer Loss (VML).** We introduce an architecture-agnostic objective that augments the base loss with a Stable-Loss controller to penalize volatility relative to an EMA baseline, reducing run-to-run variability with $1\times$ training and $1\times$ inference per deployed model.

2. **Adaptive and tuning-free control.** VML's controller self-calibrates using the ratio $\sigma_t/\sigma_{\mathrm{ref}}$, EMA baselines, and clipping to adjust its gain online. One default hyperparameter set works across models and datasets, removing per-architecture tuning.

3. **Robust gains and favorable cost–variance trade-off.** Across CIFAR-10/100 and four backbones, VML consistently lowers across-seed accuracy SD while keeping mean accuracy essentially unchanged.

Together, these findings pave the way for more reliable and trustworthy deep learning systems by addressing variability not only at the statistical level, but within the training dynamics themselves.

## 2 RELATED WORK

Robust solutions is a fundamental requirement for trustworthy deep learning, especially in domains where models are deployed under varying conditions or where reproducibility is essential for scientific credibility. Kaur et al. (2022) provide a comprehensive survey, emphasizing the need for systems that are resilient to perturbations and implementation variability in order to ensure reliable behavior. Bouthillier et al. (2019) present a critical assessment of reproducibility failures in machine learning and argue that many of these issues stem from insufficient control over experimental variability. Their study illustrates how minor implementation details, random seed choices, or system-level factors can lead to substantial performance fluctuations. Building on this, Picard (2021) specifically focuses on the role of random seeds within the broader space of randomness. He provides empirical evidence showing how seed selection can dramatically affect reported results in computer vision models. However, in the absence of a systematic strategy for seed setting, ambiguity in the results remains. Following this lack of a systematic approach, Ji et al. (2023) further analyze the effect of randomness on evaluation metrics. They recommend multi-run reporting and controlled seed strategies as a partial remedy, yet they acknowledge that determining a sufficient number of runs remains unresolved. Addressing this open question, Gundersen et al. (2023) investigate robustness against algorithmic randomness in neural network training. Crucially, they propose methodological standards requiring at least 25 repeated training runs to draw statistically sound conclusions. They argue that robustness to randomness is not a secondary technical detail but a prerequisite for trustworthy scientific and empirical claims. Several approaches have attempted to address stochastic sensitivity indirectly. Summers & Dinneen (2021) investigates nondeterminism in highly controlled environments such as ours. While they propose accelerated ensembling as a partial mitigation, this strategy does not address instability arising from fluctuations within the optimization trajectory itself; rather, it provides a statistical solution similar to ensembling. Ahmed & Lofstead (2022) propose practical strategies to manage pseudo-randomness, including consistent seeding and systematic logging of random state. They frame this as essential for improving both trustworthiness and experimental reliability. However, as Summers & Dinneen (2021) show, such control measures alone do not eliminate variability; this is functionally similar to seed fixing, it constrains the sequence of random events without reducing the underlying sensitivity to stochastic variation.

In response to these limitations, our work shifts the focus from external mitigation strategies to an internal algorithmic solution. We propose an architecture-agnostic loss function, Variance Mini-

mizer Loss (VML), that operates directly at the loss level to target the root causes of variability. While past work has explored regularization as a way to stabilize training, these methods differ fundamentally from our approach. Unlike gradient clipping (Zhang et al., 2019), which constrains parameter updates to prevent exploding gradients, VML achieves stability by smoothing the loss curve across epochs through the Stable Loss (SL) term. Compared to Sharpness-Aware Minimization (SAM) (Foret et al., 2020), which perturbs weights to seek flatter minima primarily for better generalization, VML also smooths the effective loss surface through SL but applies this smoothing at the output logit level, with the explicit objective of enhancing robustness to stochastic effects rather than only improving generalization. This approach supports the development of neural networks that behave consistently under varying stochastic conditions.

## 3 VARIANCE MINIMIZER LOSS (VML)

Training a neural network minimizes a stochastic loss over mini-batches. For a batch $\mathcal{B}_t$ at iteration $t$, the empirical loss is

$$\ell_t(\boldsymbol{\theta}) \; = \; \frac{1}{|\mathcal{B}_t|} \sum_{(\mathbf{x}_i, y_i) \in \mathcal{B}_t} \mathrm{CE}(\mathbf{f}_{\boldsymbol{\theta}}(\mathbf{x}_i), \, y_i) \,,$$

where $\mathbf{f}_{\boldsymbol{\theta}}(\mathbf{x}) \in \mathbb{R}^C$ are class logits and CE is the cross-entropy.

Since $\ell_t$ depends on initialization, sampling order, and data augmentation, the optimization trajectory may fluctuate substantially across training runs. Such fluctuations make the training dynamics more variable than what the base objective prescribes. The **Variance Minimizer Loss (VML)** augments the standard training objective with an adaptive penalty on these fluctuations. The formulation is based on three components: (i) maintaining a running baseline of the loss, (ii) quantifying deviations from this baseline, and (iii) weighting penalties according to the observed volatility. Together, these steps turn variability in the training loss into a measurable signal that can be regulated during optimization. Formally, the per-iteration training objective is

$$\mathcal{L}_t(\boldsymbol{\theta}) \; = \; \ell_t(\boldsymbol{\theta}) \; + \; w_{\mathrm{VML}} \, \mathrm{SL}_t\big(\ell_t(\boldsymbol{\theta}), \bar{\ell}_{t-1}\big), \tag{1}$$

where $\ell_t(\boldsymbol{\theta})$ is the standard cross-entropy loss, $w_{\mathrm{VML}} \geq 0$ is a global mixing weight that balances the contribution of the variance-minimizing term against the base objective, $\bar{\ell}_{t-1}$ is an exponential moving average (EMA) baseline of the batch loss, and $\mathrm{SL}_t$ denotes the stable loss penalty defined in terms of a Huber function (Gokcesu & Gokcesu, 2021) with adaptive scaling (Sec. 3.1). We denote the complete training objective by $\mathcal{L}_t$, which we refer to as the VML. This formulation treats the stable loss term not as an auxiliary diagnostic, but as an integral part of the objective, ensuring that the penalty on variability is embedded directly into the optimization process.

### 3.1 TRACKING THE BASELINE AND DEVIATION

The first step in constructing the penalty term $\mathrm{SL}_t$ in Eq. equation 1 is to establish a running baseline of the batch loss. To this end, VML maintains an exponential moving average (EMA) of $\ell_t$:

$$\bar{\ell}_t \; = \; (1 - \alpha) \, \bar{\ell}_{t-1} + \alpha \, \ell_t, \qquad \bar{\ell}_0 = \ell_0, \tag{2}$$

where $\alpha \in (0, 1)$ is a smoothing factor. This EMA baseline $\bar{\ell}_t$ represents the recent trend of the training loss against which new values are compared. The choice of $\alpha$ controls how much history is retained: a small $\alpha$ (e.g. $\alpha = 0.01$) makes the baseline very stable, averaging over a long horizon, while a larger $\alpha$ (e.g. $\alpha = 0.2$) makes it more responsive to recent changes. Thus, $\alpha$ acts as a memory parameter that determines the effective time window of the baseline. The instantaneous deviation of the current batch loss from its baseline is then

$$\delta_t = \ell_t - \bar{\ell}_{t-1}, \tag{3}$$

which is positive if $\ell_t$ exceeds the previous baseline and negative if it falls below. This deviation $\delta_t$ is the primary signal passed into the penalty function of $\mathrm{SL}_t$, with its scale later adjusted by volatility statistics (Sec. 3.2).

## 3.2 MEASURING VOLATILITY

To determine whether a deviation $\delta_t$ is significant, VML maintains a running estimate of the typical fluctuation scale. This is done with an exponential moving average of absolute deviations:

$$\sigma_t = (1 - \beta)\,\sigma_{t-1} + \beta\,|\delta_t|, \qquad \sigma_0 = 0, \tag{4}$$

where $\beta \in (0, 1)$ is a smoothing factor similar to $\alpha$ for the volatility. After a warm-up period of $T_{\text{warmup}}$ iterations, we latch a reference value

$$\sigma_{\text{ref}} = \max\{\sigma_t,\, \varepsilon\},$$

where $\varepsilon$ prevents division by zero. This $\sigma_{\text{ref}}$ represents the "typical" scale of loss fluctuations and provides a stable denominator for subsequent normalization. In later steps (Sec. 3.3), the ratio of current volatility $\sigma_t$ to this reference will determine how strongly the penalty term $\text{SL}_t$ is activated.

## 3.3 ADAPTIVE GAIN AND GATING

The penalty should not be active at all times: minor fluctuations are part of normal training dynamics and should not be over-regularized. Instead, VML is designed to respond primarily when training enters a volatile phase. To achieve this, we compare the current volatility $\sigma_t$ against a reference $\sigma_{\text{ref}}$ and compute a relative *gain* factor.

Formally,

$$\text{gain}_t = \begin{cases} \dfrac{\max\{\sigma_t - \gamma\sigma_{\text{ref}},\, 0\}}{\sigma_{\text{ref}} + \varepsilon}, & \gamma > 1, \\ \dfrac{\sigma_t}{\sigma_{\text{ref}} + \varepsilon}, & \gamma = 1, \end{cases} \tag{5}$$

where $\gamma \geq 1$ is a gating parameter and $\varepsilon$ avoids division by zero. The role of $\gamma$ is to set a tolerance threshold. When $\gamma = 1$, every fluctuation contributes proportionally to the penalty. For $\gamma > 1$, only volatility that exceeds $\gamma$ times the reference is considered; fluctuations below this level are suppressed. This design ensures that VML activates only under excess variability and remains inactive in stable regimes.

The adaptive penalty coefficient is then

$$\lambda_t^{\text{SL}} = \text{clip}(\lambda_{\text{base}} \cdot \text{gain}_t,\, \lambda_{\text{min}},\, \lambda_{\text{max}}), \tag{6}$$

which scales the strength of the stable loss regularization based purely on observed statistics. Because $\lambda_t^{\text{SL}}$ is self-adjusting, no additional manual tuning is required for the inner penalty weight; the controller adapts automatically during training.

## 3.4 DEVIATION PENALTY

Once deviations are measured, they must be penalized in a way that is both sensitive to typical fluctuations and robust against rare spikes. To achieve this, VML applies a Huber penalty to the deviation $\delta_t$:

$$\rho_\Delta(\delta_t) = \begin{cases} \frac{1}{2}\,\delta_t^2/\Delta, & |\delta_t| \leq \Delta, \\ |\delta_t| - \frac{1}{2}\Delta, & |\delta_t| > \Delta, \end{cases} \tag{7}$$

where $\Delta > 0$ is a robustness threshold. This choice provides two benefits: (i) for small deviations, the quadratic region encourages smooth convergence around the EMA baseline $\bar{\ell}_{t-1}$, and (ii) for large deviations, the linear region caps the penalty's growth, preventing instability due to rare outliers.

Recalling that $\delta_t = \ell_t - \bar{\ell}_{t-1}$ (Eq. 3), the complete stable loss term at step $t$ is

$$\text{SL}_t = \underbrace{\text{clip}\left(\lambda_{\text{base}} \cdot \frac{\max\{\sigma_t - \gamma\sigma_{\text{ref}},\, 0\}}{\sigma_{\text{ref}} + \varepsilon},\, \lambda_{\text{min}},\, \lambda_{\text{max}}\right)}_{\text{adaptive coefficient } \lambda_t^{\text{SL}}} \cdot \underbrace{\rho_\Delta(\delta_t)}_{\text{Huber penalty on deviation}}. \tag{8}$$

Together, $\delta_t$ measures deviation, $\sigma_t$ provides the volatility scale, $\lambda_t^{\text{SL}}$ adaptively adjusts the strength, and $\rho_\Delta(\delta_t)$ ensures robustness.

## 3.5 GRADIENT EFFECT

The gradient of the stable loss penalty with respect to model parameters is

$$\nabla_{\boldsymbol{\theta}}\text{SL}_t \;=\; \lambda_t^{\text{SL}}\,\psi_\Delta(\delta_t)\,\nabla_{\boldsymbol{\theta}}\ell_t, \tag{9}$$

where $\psi_\Delta(\delta_t)$ is the derivative of the Huber function $\rho_\Delta(\delta_t)$. Because $|\psi_\Delta(\delta_t)| \leq 1$, the penalty term rescales but never amplifies the gradient of the base loss. Thus, $\lambda_t^{\text{SL}}$ modulates the effective update size according to observed volatility. Combining the base loss and the stable penalty, the effective gradient used for parameter updates is

$$\nabla_{\boldsymbol{\theta}}\mathcal{L}_t(\boldsymbol{\theta}) \;=\; \left(1 + w_{\text{VML}}\,\lambda_t^{\text{SL}}\,\psi_\Delta(\delta_t)\right)\nabla_{\boldsymbol{\theta}}\ell_t, \tag{10}$$

This formulation shows that VML does not introduce an additional gradient direction but instead adaptively rescales the base gradient magnitude depending on volatility in the loss trajectory.

## 4 METHODOLOGY

To assess the effectiveness of VML, we developed a systematic experimental protocol that isolates stochastic effects and quantifies variance in both performance and training behavior.

### 4.1 DETERMINISTIC TRAINING

We began by establishing deterministic baselines through strict control of known randomness sources. To do so, we fixed random seeds across all relevant libraries and disabled nondeterministic operations at the framework level, such as cuDNN benchmarking and parallel kernel execution. This ensured that observed variability arises only from inherent stochastic effects not eliminated by seeding. All experiments were conducted under identical hardware, software, and hyperparameter settings. To evaluate sensitivity of model training to stochastic factors, we trained models across 5 different seeds $S = \{1, 2, \ldots, 5\}$. For a more rigorous analysis, we further conducted experiments with 20 seeds for the ResNet–CIFAR-10 pair. Each training run used the same architecture and optimization configuration, enabling a controlled analysis of run-to-run variability. We adopted a from-scratch training protocol, following the recommendations of Summers & Dinneen (2021), in which each model is trained independently from randomized initialization. This approach captures the full variability introduced by stochastic components in model initialization and optimization, and avoids bias from warm-started models or transfer learning. We explicitly study three of these common stochastic components: weight initialization, data shuffling, data augmentation, and the combined effect of all three. Unless stated otherwise, all results use standard augmentations: horizontal flip, padding by four pixels, and random crop. We quantified performance variability across seeds by measuring the standard deviation (SD) of test accuracy.

### 4.2 VML HYPERPARAMETERS

VML exposes a small set of scalar hyperparameters. Although the VML is adaptive and updates its internal statistics online, a few scalars must be initialized to sensible values (e.g., the warmup horizon used to form the reference statistic $\sigma_{\text{ref}}$). We therefore ran a one–factor–at–a–time ablation on the ResNet-14 / CIFAR-10 pair, sweeping each VML hyperparameter while holding all others fixed. The best setting (by mean test accuracy and SD across seeds) was then frozen and reused for all subsequent experiments (other architectures and CIFAR-100 as well). The selected configuration transfers well and does not require per-dataset recalibration, consistent with VML's adaptive nature. All other training hyperparameters (optimizer, schedule, batch size, etc.) are fixed and shared between conditions. See Appendix **??**, Table 5 for SL hyperparameters.

### 4.3 EXPERIMENTAL SETUP

Our experiments are conducted across multiple pairs of models and datasets for image classification. We utilize the CIFAR-10 and CIFAR-100 datasets. We evaluate four convolutional neural networks: ResNet-14, MobileNet-V2, VGG-16, and ShuffleNet-V2. These models cover a broad range of design strategies. ResNet uses residual connections to enable deeper training. MobileNet-V2 is

lightweight and optimized for efficiency. VGG-16 is a classical deep model with uniform structure. ShuffleNet-V2 is designed for speed and low resource cost. By selecting these diverse architectures, we test the generality of our method across both heavy and lightweight models. Training follows a cosine decay learning rate schedule with an initial peak of 0.40, a batch size of 512, momentum set to 0.9, and a weight decay of $5 \times 10^{-4}$. All experiments are conducted using PyTorch Paszke et al. (2019) on a compute environment with 64 CPU cores, 512 GB of RAM, and NVIDIA A100 GPU.

# 5 RESULTS AND DISCUSSION

This section presents a comprehensive evaluation of the VML across various settings of image classification tasks. Overall, we executed 255 fixed, identical training sessions, along with a SL internal parameter exploration, totaling approximately 230 hours of GPU time.

## 5.1 DIAGNOSING VML REACTIVITY TO LOSS DYNAMICS

We analyze to what extend the SL controller in VML (Sec. 3) is active and temporally coupled to training dynamics. As in §3, $t$ indexes *iterations*. We introduce an epoch index $k$ and denote by $\mathcal{I}_k$ the iterations inside epoch $k$. We aggregate the per-iteration controller coefficient from Eq. equation 6 into an epoch-level signal

$$\lambda_k \ := \ \frac{1}{|\mathcal{I}_k|} \sum_{t \in \mathcal{I}_k} \lambda_t^{\mathrm{SL}},$$

and denote the test loss evaluated at the end of epoch $k$ by $\ell_k^{\mathrm{test}}$. (All series are averaged across seeds; conclusions are unchanged if computed per seed.) To focus on changes rather than levels we use first differences

$$\Delta\lambda_k := \lambda_k - \lambda_{k-1}, \qquad \Delta\ell_k^{\mathrm{test}} := \ell_k^{\mathrm{test}} - \ell_{k-1}^{\mathrm{test}}.$$

Because $\Delta\lambda_k$ is numerically small, we z-score each sequence for the scatter plot: $z(\Delta x_k) := (\Delta x_k - \mu_{\Delta x})/\sigma_{\Delta x}$. This removes units and makes the effect size comparable across axes.

**Cross–correlation function (CCF):** We quantify timing via the Pearson cross–correlation

$$r_\tau \ := \ \mathrm{corr}\big(\Delta\lambda_k, \ \Delta\ell_{k+\tau}^{\mathrm{test}}\big), \qquad \tau \in [-10, 10],$$

where a *positive* lag $\tau > 0$ means changes in the controller $\Delta\lambda_k$ *lead* changes in test loss, and $\tau < 0$ means they *lag*. (Here $\mathrm{corr}(\cdot, \cdot)$ is the Pearson correlation coefficient $r \in [-1, 1]$, with $r>0$ indicating positive linear association and $r<0$ negative.) On CIFAR10/ResNet-14, the CCF (right panel of Fig. 1) peaks at $\tau^\star = +1$ with $r_{\tau^\star} \approx 0.44$, consistent with the SL design in §3: the controller $\lambda_t^{\mathrm{SL}}$ reacts to volatility estimated from $\delta_t = \ell_t - \bar{\ell}_{t-1}$ and $\sigma_t$ (Eqs. 3–4), and this adjustment influences the *next* epoch's outcome.

**Standardized scatter:** The left panel of Fig. 1 plots one blue marker per epoch $k$, whose coordinates are the paired standardized changes $\big(z(\Delta\lambda_k), z(\Delta\ell_{k+1}^{\mathrm{test}})\big)$. Thus each dot shows how a change in the controller during epoch $k$ relates to the change in test loss in epoch $k+1$. We overlay the least-squares best-fit line to summarize the trend. Because both axes are standardized, the slope of this line equals the Pearson correlation $r$. We obtain $r \approx 0.435$ with two-sided $p \approx 1.6 \times 10^{-10}$, indicating a statistically significant, moderate positive association: when the controller strengthens ($\Delta\lambda_k > 0$) in response to volatility, the subsequent change in test loss tends to move in the same direction.[1]

The $\tau = +1$ CCF peak together with the significant standardized association confirms that VML's controller $\lambda_t^{\mathrm{SL}}$ is engaged in volatile phases and its adjustments are reflected in the following epoch's test loss. These diagnostics complement the across-seed variance results by revealing how stabilization operates during training.

---

[1]This diagnostic targets *timing and coupling*; effects on the mean and variance across seeds are reported separately.

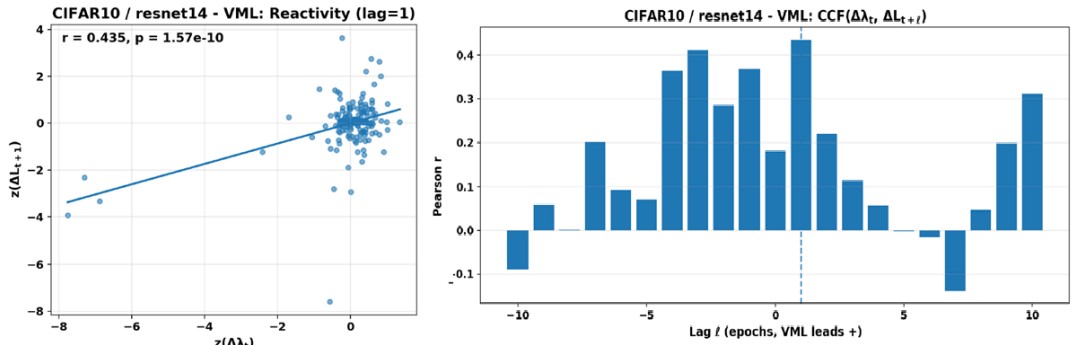

Figure 1: **VML reactivity and temporal coupling (CIFAR-10 / ResNet-14).** *Left:* Standardized scatter with one blue dot per epoch $k$, showing $z(\Delta\lambda_k)$ on the x-axis and $z(\Delta\ell_{k+1}^{\text{test}})$ on the y-axis; the least-squares line (slope$= r$) summarizes the trend. We find $r \approx 0.435$ ($p \approx 1.6 \times 10^{-10}$). *Right:* Cross–correlation function $r_\tau = \text{corr}(\Delta\lambda_k, \Delta\ell_{k+\tau}^{\text{test}})$ for $\tau \in [-10, 10]$; the peak at $\tau = +1$ indicates controller updates precede changes in test loss by one epoch.

Table 1: Mean accuracy and variability across various stochastic sources. Var. reduction is computed relative to the Base SD within the same setting.

| Sources | Base Acc. (%) $\pm$ SD | VML Acc. (%) $\pm$ SD | Var. Red. (%) |
|---|---|---|---|
| All | $94.95 \pm 0.169$ | $94.88 \pm 0.112$ | 33.3 |
| Init | $94.90 \pm 0.194$ | $94.91 \pm 0.112$ | 42.4 |
| Aug | $94.87 \pm 0.167$ | $94.82 \pm 0.043$ | 74.0 |
| Shuf | $94.82 \pm 0.159$ | $94.88 \pm 0.106$ | 33.1 |

## 5.2 VARIABILITY REDUCTION ACROSS STOCHASTIC SOURCES

To identify where VML reduces run-to-run variability, we isolate each source of stochasticity in turn. We evaluate them: **Init** (only weight initialization varies; augmentation and data shuffling random number generators are fixed), **Aug** (only augmentation randomness varies), **Shuf** (only data-loader shuffling varies), and **All** (all three vary simultaneously). For each regime we report the mean final test accuracy and the across-run standard deviation of accuracy ("Acc. SD"), computed over repeated seeds. We summarize variability reduction as

$$\text{Var. Red.} = 100 \times \left(1 - \frac{\text{SD}_{\text{VML}}}{\text{SD}_{\text{Base}}}\right),$$

that is, the percentage drop in across-run spread when replacing the base objective with VML. Table 1 shows a consistent reduction in variability for VML in every regime while leaving average accuracy essentially unchanged (differences are within typical noise). The largest variance reduction occurs when augmentation is the only active randomness (**Aug**), which is also the setting where per-iteration losses fluctuate most due to instance-level transforms; in this case VML's SL controller down-weights volatile steps, leading to tighter outcomes. When only initialization varies (**Init**) or only data shuffling varies (**Shuf**), VML still reduces spread, indicating lower sensitivity to the starting point and to minibatch ordering. With all sources active simultaneously (**All**), variability remains lower under VML, demonstrating that the effect is robust when randomness sources act together.

## 5.3 VARIABILITY REDUCTION ACROSS MODELS AND DATASETS

We assess VML as a drop-in objective on CIFAR-10/100 across four architectures (ResNet-14, VGG-16, ShuffleNet-V2, MobileNetV2). For each dataset–model pair, Table 2 reports the mean final test accuracy, the across-run standard deviation of accuracy ("Acc. SD"), and the within-pair variability reduction. Across all models and both datasets, VML consistently lowers Acc. SD while keeping average accuracy essentially unchanged. Reductions are most pronounced for lightweight/mobile backbones, which are typically more sensitive to stochasticity in data order and

Table 2: Reduction of variability through the introduction of VML across CIFAR-10/100 and multiple architectures. Each dataset–model pair shows Base and VML side by side for easy comparison.

| Dataset | Model | Base Acc. (%) ± SD | VML Acc. (%) ± SD | Var. Red. (%) |
|---|---|---|---|---|
| CIFAR-10 | ResNet-14 | 94.95 ± 0.168 | 94.88 ± 0.112 | 33.3 |
| | VGG-16 | 93.74 ± 0.140 | 93.89 ± 0.098 | 29.7 |
| | ShuffleNet-V2 | 91.08 ± 0.217 | 91.01 ± 0.056 | 74.4 |
| | MobileNetV2 | 92.03 ± 0.232 | 91.98 ± 0.147 | 36.3 |
| CIFAR-100 | ResNet-14 | 75.66 ± 0.304 | 75.68 ± 0.112 | 63.3 |
| | VGG-16 | 74.17 ± 0.384 | 73.93 ± 0.200 | 48.1 |
| | ShuffleNet-V2 | 67.97 ± 0.465 | 68.11 ± 0.279 | 40.0 |
| | MobileNetV2 | 69.59 ± 0.288 | 69.52 ± 0.079 | 72.7 |

Table 3: Variability reduction across seed-group sizes on CIFAR-10 / ResNet-14. Each cell reports mean accuracy ± SD across runs; Var. Red. is the percentage drop in SD from Base to VML within the same group size.

| Group Size | Base Acc. (%) ± SD | VML Acc. (%) ± SD | Var. Red. (%) |
|---|---|---|---|
| 5 | 94.95 ± 0.168 | 94.88 ± 0.112 | 33.3 |
| 10 | 94.89 ± 0.140 | 94.85 ± 0.114 | 18.0 |
| 15 | 94.94 ± 0.132 | 94.88 ± 0.118 | 10.5 |
| 20 | 94.89 ± 0.129 | 94.85 ± 0.125 | 3.0 |

augmentation; standard backbones also benefit, albeit to a lesser extent when the baseline variability is already small (leaving less room to improve under a ratio metric). The same qualitative pattern appears on CIFAR-100, indicating that VML's effect is not tied to a particular dataset difficulty. Overall, the variability reduction spans roughly 33% to 74% across the dataset–model pairs in Table 2. It is worth noticing that we do not target accuracy-optimal tuning here, Achieving the highest possible accuracy is not the goal of this analysis. While we could also pursue accuracy-optimal settings, this would substantially increase computational cost and time, since our analysis requires multiple independent training runs with different seeds. This design emphasizes stability under repeated training rather than one-off peak performance.

## 5.4 SAMPLE-BASED EVALUATION OF VML ROBUSTNESS

To test whether VML's stability is robust to the choice and number of seeds available at evaluation time, we use a fixed pool of 20 CIFAR-10/ResNet-14 runs per training regime (Base vs. VML) and form seed groups of sizes 5, 10, 15, and 20. For group sizes 5–15 we repeatedly subsample seed sets from the pool (fixed-size resampling; the full set is used for size 20), compute the group's mean accuracy and across-run standard deviation ("Acc. SD"), and then summarize these statistics per regime. As reported in Table 3, VML reduces Acc. SD for every group size while leaving mean accuracy essentially unchanged. The variability reduction is largest for small groups and tapers as the group size increases, which is expected because averaging over more independent runs already dampens seed noise in the baseline. At the full pool size (20 runs), the Base and VML estimates become almost indistinguishable, consistent with prior guidance that drawing seed-insensitive conclusions typically requires on the order of 25 independent trainings, since groups of "good" and "bad" seeds tend to average out at that scale (Gundersen et al., 2023). Overall, these results show that VML's benefit is not tied to a particular seed choice and remains useful in practical settings where only a limited number of runs can be afforded.

## 5.5 COMPARISON TO EXISTING METHODOLOGIES

Table 4 compares VML with full ensembling, accelerated (snapshot/EMA) ensembling (Wen et al., 2020), and test-time data augmentation (TTA) (Szegedy et al., 2014) on CIFAR-10/ResNet-14. Full ensembling gives the largest SD drop but costs $20\times$ training. Accelerated ensembling keeps training at $1\times$ (via EMA/snapshots) but typically reduces less. TTA stabilizes predictions by averaging $K$ stochastic test-time views, incurring about $K\times$ inference. In contrast, **VML** is a single-model, single-view method with $1\times$ training and $1\times$ inference; it improves stability by directly shaping

Table 4: Method comparison on CIFAR 10 with ResNet-14. Training cost is measured relative to a single-model run. Variability reduction is the percentage drop in across-seed accuracy SD relative to the single-model baseline.

| Method | Training Cost | Model | Dataset | Variability Reduction (%) |
|---|---|---|---|---|
| Ensemble ($N$=20)[1] | $20\times$ | ResNet-14 | CIFAR 10 | 58 |
| Accelerated Ensemble[2] | $1\times$ | ResNet-14 | CIFAR 10 | 27.0 |
| Test-time Data Aug. ($K$ views)[3] | $K\times$ (inference) | ResNet-14 | CIFAR 10 | 30.7 |
| VML (Ours) | $1\times$ | ResNet-14 | CIFAR 10 | **33.3** |

optimization through the SL controller and is architecture-agnostic. Among $1\times$-training baselines, VML offers the best cost vs. variance trade-off, matching or exceeding their variability reduction without extra runs or multi-view inference. Complementing the SD results, the seed effect size (top-10 vs. bottom-10 seeds) on CIFAR-10/ResNet-14 drops from 0.85 (prior work) to 0.63 with VML, narrowing the gap between "good" and "bad" seeds under the same budget.

## 6 CONCLUSION AND FUTURE WORK

We introduced the **Variance Minimizer Loss (VML)**, a simple, architecture-agnostic objective that reduces run-to-run variability by modulating a stable-loss coefficient $\lambda_t$ in response to volatility in the training signal. On CIFAR-10/100 with ResNet-14, VGG-16, ShuffleNet-V2, and MobileNetV2, VML consistently lowers across-seed accuracy SD while keeping mean accuracy essentially unchanged, with the largest gains in regimes that induce high step-to-step variability (e.g., heavy augmentation). Our diagnostics show that controller updates lead changes in test loss by one epoch (a cross-correlation peak at $+1$) and that standardized changes $z(\Delta\lambda_t)$ co-move with $z(\Delta L_{t+1})$, confirming that the controller is most active when the signal is unstable. Compared to common variance-reduction baselines, VML offers a favorable cost–variance trade-off: unlike full or accelerated ensembling and test-time augmentation, it keeps both training and inference at $1\times$. Beyond SD, the seed effect size on CIFAR-10/ResNet-14 (gap between the average of the top-10 and bottom-10 seeds) drops from 0.85 to 0.63 with VML, narrowing the spread between "good" and "bad" seeds under the same budget.

**Limitations.** Estimating variability requires many independent trainings, which makes our analysis computationally expensive; as a result, we have not yet validated the approach at ImageNet scale. We also fixed the controller hyperparameters (e.g., EMA windows, gating threshold $\gamma$, and clipping bounds) across all experiments. A systematic ablation of these settings across different model-dataset pairs may yield further gains and clarify sensitivity. Addressing these limitations, namely scaling to larger benchmarks and exploring the hyperparameter design space, will require additional experimentation and is part of our planned future work.

---

[1]Training and storing $N$ independently trained models; cost scales linearly with $N$.

[2]E.g., snapshot ensembles, SWA/EMA checkpoints; multiple predictions from one or a few training schedules.

[3]Average $K$ stochastic test-time transforms per image; no extra training, but $K\times$ inference cost.

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

## A  FURTHER DETAILS ON FORMULATION OF VML

This appendix collects the full mathematical details for the Stable Loss (SL) and the Variance Penalty Loss (VPL) used in the Variance Minimizer Loss (VML). The main paper presents the streamlined formulation and intuition.

### A.1  NOTATION SNAPSHOT

We follow the main text: $\boldsymbol{\theta} \in \Theta \subseteq \mathbb{R}^d$ are parameters, $\mathbf{x} \in \mathcal{X}$ inputs, $y \in \{1, \ldots, C\}$ labels, $\mathbf{f}_{\boldsymbol{\theta}}(\mathbf{x}) \in \mathbb{R}^C$ logits (pre-softmax), $\ell_t(\boldsymbol{\theta})$ the mini-batch loss at step $t$. For a class $c$, $S_c = \{i : y_i = c\}$ and $m_c = |S_c|$, $\bar{\mathbf{f}}_c := m_c^{-1} \sum_{i \in S_c} \mathbf{f}_{\boldsymbol{\theta}}(\mathbf{x}_i)$ with $c$-th component $\bar{f}_{c,c}$. The eligible set is $\mathcal{C}_t = \{c : m_c \geq 2\}$.

### A.2  STABLE LOSS (SL): EXTENDED DETAILS

**Huber penalty.** SL applies a Huber penalty to the deviation $\delta_t = \ell_t(\boldsymbol{\theta}) - \bar{\ell}_{t-1}$:

$$\rho_\Delta(u) = \begin{cases} \frac{1}{2} u^2/\Delta, & |u| \leq \Delta, \\ |u| - \frac{1}{2}\Delta, & |u| > \Delta, \end{cases} \tag{11}$$

with soft threshold $\Delta > 0$. This is quadratic near 0 and linear for large $|u|$.

**Threshold modes.** We use either a fixed absolute threshold or a scale-free fractional one:

$$\Delta = \delta_{\text{abs}} > 0 \quad \text{or} \quad \Delta = \delta_{\text{frac}} \cdot s_t, \;\; s_t \in \{\sigma_t, \sigma_{\text{ref}}\},$$

where $\sigma_t$ is an EMA of $|\delta_t|$ and $\sigma_{\text{ref}}$ a latched reference.

**SL gradient.** The derivative of equation 11 is

$$\psi_\Delta(u) = \partial_u \rho_\Delta(u) = \begin{cases} u/\Delta, & |u| \leq \Delta, \\ \text{sign}(u), & |u| > \Delta, \end{cases}$$

so the SL contribution to the gradient (controller treated as constant) is

$$\nabla_{\boldsymbol{\theta}} \text{SL}_t = \lambda_t^{\text{SL}} \, \psi_\Delta(\delta_t) \, \nabla_{\boldsymbol{\theta}} \ell_t(\boldsymbol{\theta}),$$

with $|\psi_\Delta(\delta_t)| \leq 1$.

**Volatility gating.** With gate $\gamma > 1$ and reference $d_t = \max\{\sigma_{\mathrm{ref}}, \varepsilon\}$, the gain in the main text is zero whenever $\sigma_t \le \gamma d_t$, keeping SL inactive in calm regimes. The adaptive weight is $\lambda_t^{\mathrm{SL}} = \mathrm{clip}(\lambda_{\mathrm{base}}^{\mathrm{SL}} \cdot \mathrm{gain}_t, \lambda_{\min}^{\mathrm{SL}}, \lambda_{\max}^{\mathrm{SL}})$.

## A.3 Compact algorithmic summary

---

**Algorithm 1** VML training step (base loss + Stable Loss; controller updates are stop-grad)

---

1: **Inputs:** batch $\mathcal{B}_t = \{(\mathbf{x}_i, y_i)\}_{i=1}^{N_t}$; network $f_{\boldsymbol{\theta}}$; controller state $\{\bar{\ell}_{t-1}, \sigma_{t-1}, \sigma_{\mathrm{ref}}\}$
2: **Hyperparams:** EMA factors $\alpha, \beta$; gating $\gamma$; Huber threshold $\Delta$; warmup steps $W$; base gain $\lambda_{\mathrm{base}}$; clip $[\lambda_{\min}, \lambda_{\max}]$; mix $w_{\mathrm{VML}}$; $\varepsilon$
3: **Forward & base loss**
4: $\quad z_i \leftarrow f_{\boldsymbol{\theta}}(\mathbf{x}_i); \quad \ell_t \leftarrow \frac{1}{N_t} \sum_i \mathrm{CE}(z_i, y_i)$
5: **Update SL statistics (stop-grad)**
6: $\quad \bar{\ell}_t \leftarrow (1 - \alpha)\,\bar{\ell}_{t-1} + \alpha\,\mathrm{sg}(\ell_t)$          (EMA baseline; cf. Eq. equation 2)
7: $\quad \delta_t \leftarrow \mathrm{sg}(\ell_t) - \bar{\ell}_{t-1}$          (deviation; Eq. equation 3)
8: $\quad \sigma_t \leftarrow (1 - \beta)\,\sigma_{t-1} + \beta\,|\delta_t|$          (volatility EMA; Eq. equation 4)
9: $\quad$ **if** $t = W$ **then** $\sigma_{\mathrm{ref}} \leftarrow \max\{\sigma_t, \varepsilon\}$          (latch reference)
10: **Compute adaptive SL gain and weight**
11: $\quad$ **if** $\gamma > 1$ **then** $\mathrm{gain}_t \leftarrow \dfrac{\max\{\sigma_t - \gamma\,\sigma_{\mathrm{ref}}, 0\}}{\sigma_{\mathrm{ref}} + \varepsilon}$ **else** $\mathrm{gain}_t \leftarrow \dfrac{\sigma_t}{\sigma_{\mathrm{ref}} + \varepsilon}$    (Eq. equation 5)
12: $\quad \lambda_t^{\mathrm{SL}} \leftarrow \mathrm{clip}(\lambda_{\mathrm{base}} \cdot \mathrm{gain}_t, \lambda_{\min}, \lambda_{\max})$          (Eq. equation 6)
13: **Stable-Loss penalty and total objective**
14: $\quad \rho_\Delta(\delta_t) \leftarrow \begin{cases} \frac{1}{2}\,\delta_t^2/\Delta, & |\delta_t| \le \Delta \\ |\delta_t| - \frac{1}{2}\Delta, & |\delta_t| > \Delta \end{cases}$          (Huber; Eq. equation 7)
15: $\quad \mathrm{SL}_t \leftarrow \lambda_t^{\mathrm{SL}} \cdot \rho_\Delta(\delta_t)$
16: $\quad \mathcal{L}_t \leftarrow \ell_t + w_{\mathrm{VML}}\,\mathrm{SL}_t$          (total objective; Eq. equation 1)
17: **Backprop & update**    $\boldsymbol{\theta} \leftarrow \mathrm{SGD}/\mathrm{Adam}(\nabla_{\boldsymbol{\theta}} \mathcal{L}_t)$
18: **Note:** $\mathrm{sg}$ denotes stop-gradient; no gradients flow through $\bar{\ell}_t, \sigma_t, \sigma_{\mathrm{ref}}, \mathrm{gain}_t,$ or $\lambda_t^{\mathrm{SL}}$.

---

# B Stable-Loss Internals

Table 5: Stable-Loss (SL) hyperparameters, implementation names, roles, and values. Values were selected via a ResNet-14/CIFAR-10 ablation and reused unchanged across all experiments.

| Parameter (Sec. 3) | Implementation arg | Role / intuition | Chosen |
|---|---|---|---|
| $\lambda_{\mathrm{base}}$ | lambda_base | Base SL gain; after warmup it scales as $\lambda_t = \lambda_{\mathrm{base}} \cdot \sigma_t/\sigma_{\mathrm{ref}}$ (clipped). | **0.10** |
| $\alpha$ | alpha | EMA blend for $\bar{L}_t$; smaller $\Rightarrow$ slower, stabler baseline. | **0.05** |
| $\beta$ | beta | EMA blend for volatility $\sigma_t = \mathrm{EMA}_\beta(|\Delta_t|)$; sets responsiveness to loss deviations. | **0.20** |
| $\delta$ | delta | Huber threshold in $\mathcal{H}_\delta(\Delta_t)$; sets quadratic→linear transition for robustness. | **0.20** |
| $W$ (warmup steps) | warmup_steps | Steps used to capture the reference $\sigma_{\mathrm{ref}}$; SL gain adapts relative to this reference thereafter. | **200** |
| $[\lambda_{\min}, \lambda_{\max}]$ | lambda_min / lambda_max | Safety bounds on $\lambda_t$ to prevent extreme gains. | **[0.0, 2.0]** |
| $\varepsilon$ | eps | Numerical stabilizer in divisions and clamps. | $10^{-8}$ |

## B.1 Seed–group view via kernel density estimates

To visualize across–seed variability, Fig. 2 plots kernel density estimates (KDEs) of the final test accuracy for the CIFAR 10 / ResNet-14 runs under *Base* and *VML*. The KDEs are normalized (area $= 1$), so taller peaks indicate smaller dispersion, while horizontal spread reflects variability. With $n=5$ seeds (left), both distributions are relatively broad and sampling noise is evident; nevertheless, VML already concentrates mass more tightly around its mode. With $n=20$ seeds (right), the picture

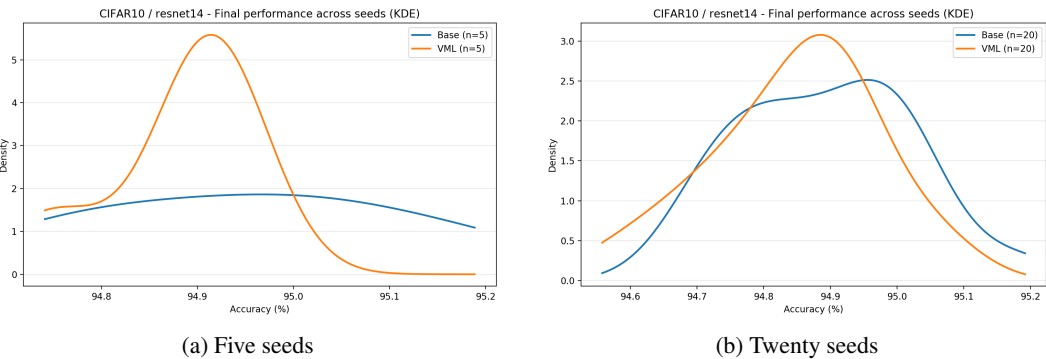

(a) Five seeds          (b) Twenty seeds

Figure 2: **Across–seed distribution of final accuracy** on CIFAR 10 / ResNet-14. KDE curves (area = 1) show similar central tendencies for Base and VML but reduced spread under VML, especially with more repeats, consistent with Table 3.

stabilizes: the Base and VML modes remain close (means essentially unchanged), but the VML curve is visibly narrower with lighter tails, mirroring the lower accuracy SD reported in Table 3. The contrast between the two panels also illustrates why small seed groups can be misleading: estimates with five repeats are noisy, whereas twenty repeats yield a clearer—and more reliable—gap in dispersion in favor of VML.

## B.2 ABLATION OF THE SL CONTROLLER HYPERPARAMETERS

We ablate the Stable–Loss (SL) controller on CIFAR 10 / ResNet-14 by varying one knob at a time while fixing the others at their defaults (Table 5): **SLA** ($\alpha$, EMA for the loss baseline $\bar{L}_t$), **SLB** ($\beta$, EMA for the volatility $\sigma_t$), **SLD** ($\delta$, Huber transition), **SLW** ($W$, warm-up steps for $\sigma_{\mathrm{ref}}$), and **SW** ($\lambda_{\mathrm{base}}$, base gain for $\lambda_t$). Each panel in Fig. 3 shows a Pareto-style scatter of *across-seed standard deviation* (x, lower is better) versus *mean accuracy* (y, higher is better); points are annotated with the setting (e.g., SLA005=$\alpha$=0.05).

**Findings.** (1) **Baseline smoothing** ($\alpha$) exhibits a clear sweet-spot: very small $\alpha$ tracks too slowly (higher variance), and very large $\alpha$ over-reacts; $\alpha \approx 0.05$ gives the best stability–accuracy balance. (2) **Volatility EMA** ($\beta$) benefits from faster adaptation: larger $\beta$ reduces variance more consistently. (3) **Huber threshold** ($\delta$) is best at a moderate value: too small makes SL effectively linear (noisy), while too large delays activation. (4) **Warm-up** ($W$): a longer warm-up (e.g., $W$=200) stabilizes $\sigma_{\mathrm{ref}}$ and lowers variance with only minor movement in the mean. (5) **Base gain** ($\lambda_{\mathrm{base}}$) shows a U-shaped trade-off: very small under-activates SL (little reduction), very large over-regularizes (hurting stability and/or mean). A mid-range value ($\lambda_{\mathrm{base}} \approx 0.10$) sits near the knee.

These trends support the defaults used throughout the paper: they lie close to the low-variance region while preserving accuracy, and we reuse the same settings across models and datasets without per-architecture tuning.[2]

---

[2]Runs were grouped with `--hp_tag_prefix` (e.g., SLA, SLB, SLD, SLW, SW); example: `--dataset cifar10 --arch resnet14 --out analysis/sl_internal/sl_sw_sweep --hp_tag_prefix SW`.

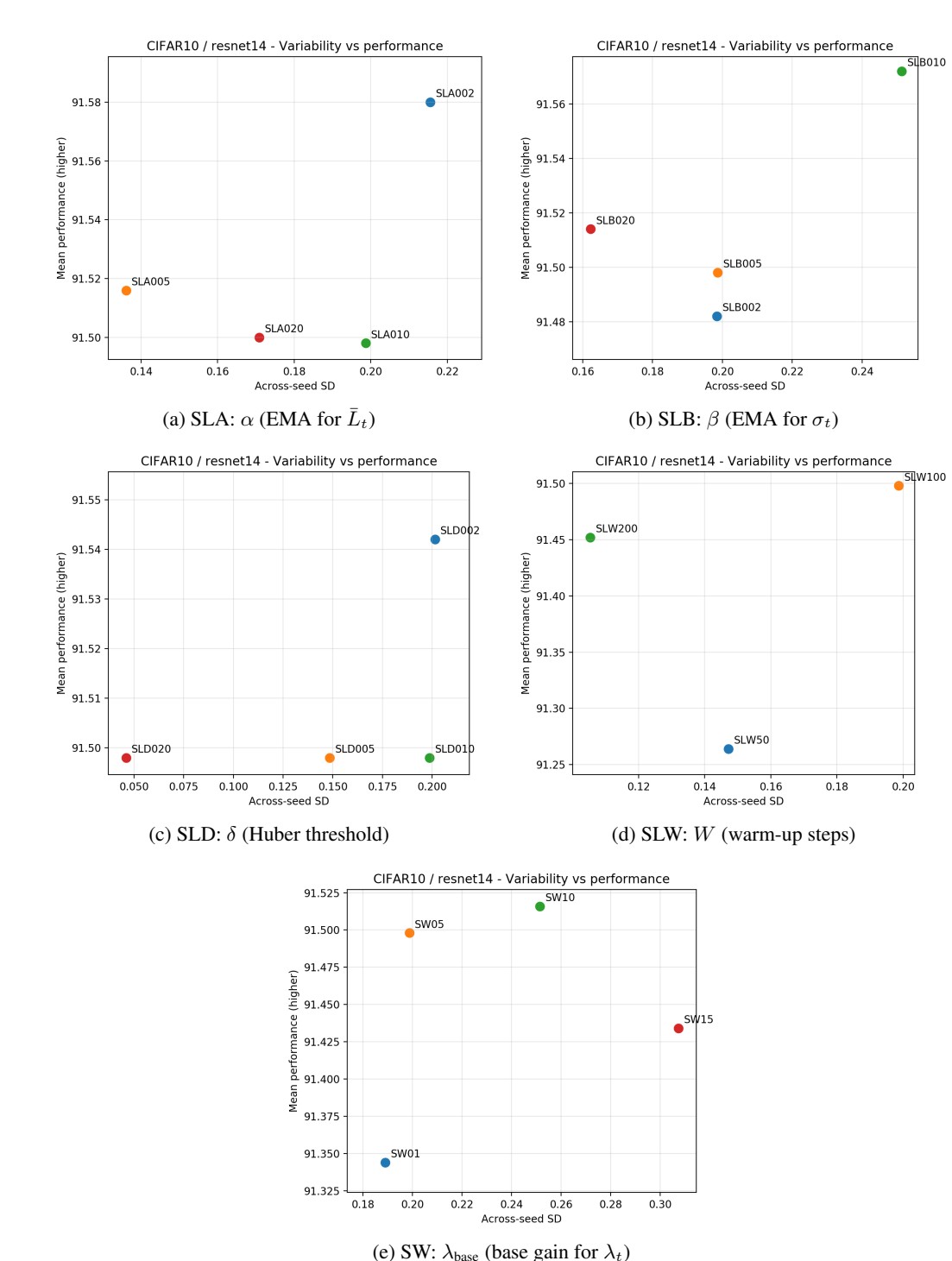

Figure 3: **SL ablations (Pareto view).** Each point shows mean accuracy (y) vs. across-seed SD (x) for one setting, annotated with its tag (e.g., `SLA005` $\Rightarrow$ $\alpha$=0.05). Lower-left is better; the recommended defaults (Table 5) sit near the low-variance knee without sacrificing accuracy.