# OpenReview forum: "Towards Robust Neural Networks via Variance Minimizer Loss"
_ICLR.cc/2026/Conference — ICLR 2026 Conference Withdrawn Submission_

### Official Review · Reviewer_m4yg · 2025-10-19

**Soundness:** 2
**Presentation:** 2
**Contribution:** 1
**Rating:** 4
**Confidence:** 3

**Summary:**

This paper aims to address the issue of poor reproducibility arising from variations in random seeds. The authors propose to rescale the gradient magnitude based on the estimated variance of the current optimization step, which is computed by comparing the current loss value with an exponentially moving average (EMA) of past losses. The proposed method is evaluated on the CIFAR-10 and CIFAR-100 datasets using different CNN architectures.

**Strengths:**

1. Clear motivation. The authors aim to address the issue of reproducibility in current deep learning algorithms, which arises from the inherent randomness in each training run.
2. Clear writing. The paper is well written and easy to follow.

**Weaknesses:**

1. Limited novelty. The proposed method demonstrates limited originality. In essence, it rescales the loss gradient based on an estimated variance computed from an EMA-updated loss value. Moreover, the paper lacks theoretical analysis regarding the convergence of the proposed method, which is crucial given that the main contribution lies in modifying the optimization algorithm.
2. Limited experiments. Although the authors claim that the proposed method is architecture-agnostic, the experiments are conducted only with CNN-based models, while a large family of Transformer architectures is not considered. Moreover, the evaluation is limited to two small-scale datasets, CIFAR-10 and CIFAR-100, without validation on more complex datasets such as ImageNet. Since larger datasets typically introduce greater variance in SGD updates, evaluating the proposed method on such datasets would be necessary to demonstrate its capability in mitigating large variance.
3. Limited comparison. The paper does not include comparisons with established optimization algorithms known to mitigate the variance of neural network updates, such as Adam [1] and SAM [2]. Including such comparisons would better highlight the benefits and unique contributions of the proposed method.
4. Typo: Line 263, Appendix ??

Reference

[1] Adam: A Method for Stochastic Optimization, ICLR, 2015

[2] Sharpness-aware Minimization for Efficiently Improving Generalization, ICLR, 2021

**Questions:**

See weaknesses.

---

### Official Review · Reviewer_pDiZ · 2025-10-29

**Soundness:** 2
**Presentation:** 2
**Contribution:** 2
**Rating:** 2
**Confidence:** 3

**Summary:**

This paper proposes a variance minimizer loss (VML) that augments the base loss with a stable-loss controller. The controller tracks the exponential-moving-average of the loss, measures deviations, and adaptively penalizes volatility using a Huber-style term. Experimental results shows that the standard deviation of the proposed loss is reduced, while the test accuracy maintains.

**Strengths:**

S1. Clear motivation: addresses an important but under-quantified form of instability.

S2. Sound ablations: controlled experiments isolating initialization, augmentation, and shuffling randomness.

S3. Transparent methodology: deterministic baselines, 20-seed evaluation, cross-correlation diagnostics linking the controller’s dynamics to loss behavior.

**Weaknesses:**

W1. Empirical scale: evaluations are limited to CIFAR-level benchmarks; generalization to ImageNet-scale or transformer architectures remains untested.

W2. Several design choice in VML is not carefully designed. For example, why we use huber type loss? Why not just square loss? Why there has to be an adaptive coefficient? And also the hyper-parameters $\lambda_{base}$, $\lambda_{min}$, $\lambda_{max}$ make the proposed loss much more complicated than a vanilla loss.

W3. Lack of theoretical support. The loss is time varying. It is not clear that, optimizing a changing loss will converge. For example, in the simpler logistic regression, where the vanilla loss is convex, can the author show that VML leads to convergence?

**Questions:**

See above

---

### Official Review · Reviewer_hUdY · 2025-10-31

**Soundness:** 3
**Presentation:** 1
**Contribution:** 3
**Rating:** 4
**Confidence:** 4

**Summary:**

The paper addresses the issue of seed-dependent variability in deep learning model performance, which challenges the robustness and fairness of reported results. It introduces Variance Minimizer Loss (VML), an adaptive and architecture-agnostic objective designed to reduce stochastic fluctuations during training. Experiments are conducted on CIFAR-10/100 using four ConvNet architectures, without incurring additional computational overhead.

**Strengths:**

**S1.** I believe that exploring variance reduction techniques, particularly those that are architecture-agnostic, represents a valuable research direction. In my opinion, the approach proposed in this paper is indeed interesting and has potential.

**S2.** I appreciate the analysis presented in Table 4. It is interesting to see that VML is less expensive than other variance reduction approaches, such as ensembling and TTA.

**Weaknesses:**

**W1.** I find the main motivation of the paper relatively weak in its current form. The claim that models yield substantially different results across runs under different conditions is only partially valid, especially given the reported coefficients of variation, which are extremely small (e.g., 0.001 in Table 1). The framing of the motivation could be made considerably stronger by focusing on more meaningful sources of variability. For example, investigating variability in small-sample learning scenarios, such as training on different random subsets of limited data (e.g., 1% of the CIFAR10 training set), could provide a more compelling and practically relevant demonstration of the proposed method’s value.

**W2.** The description of the methodology is not very easy to follow, considering that there are multiple parameter definitions, including EMAs, etc. It would be very helpful to include a figure (iterations vs. [VARIABLE]) illustrating the evolution of, for example, the default/EMA loss, volatility, and other defined variables, to provide a more intuitive picture while reading the method description.

**W3.** The authors train several ConvNets, but are missing evaluations on ViTs. Considering the much higher stochastic nature of the training loss for this architecture's type, I wonder whether the approach works there as well.

**W4.** Since VLM needs several HPs, and such HPs have been picked by sweeping on CIFAR 10, but tested only on CIFAR100 as a “different” dataset, how wonder how they would behave on datasets that have more spiky/different training loss evolutions.

**W5.** Figure 1 is blurry. I recommend using more suitable image formats.

**Questions:**

**Q1.** Why are you not using the coefficient of variation to measure volatility in the evaluation part?

In summary, I found the paper interesting, but in its current form, it is more suitable for a workshop than a full conference paper. It would need some improvement in the motivation framing, experimental contribution, and presentation. Welcoming feedback from the authors during the discussion period.

---

### Official Review · Reviewer_FcHt · 2025-11-01

**Soundness:** 3
**Presentation:** 2
**Contribution:** 3
**Rating:** 4
**Confidence:** 3

**Summary:**

The paper proposes Variance Minimizer Loss (VML) to augment the base loss with a Stable Loss (SL) term to reduce run-to-run variance stemming from randomness in initialization, data shuffling, and augmentation. Concretely, the method maintains an EMA baseline of the batch loss, measures deviations from that baseline, estimates volatility via an EMA of absolute deviations, and uses a gated, clipped controller to adapt the penalty strength. The penalty uses a Huber form for robustness. Empirically, on CIFAR-10/100 and four architectures (ResNet-14, VGG-16, MobileNetV2, ShuffleNet-V2), VML reduces across-seed accuracy standard deviations by ~33–75% with almost no change in mean accuracy and, more importantly, no added training/inference cost.

**Strengths:**

1. The paper focuses squarely on across-seed variance as an optimization target. The controller (EMA baseline, EMA volatility, gated/clipped gain) is a simple design and easy to implement.
2. The proposed method controls distinct randomness sources (init/aug/shuffle) and reports their individual contributions to variance reduction.
3. The proposed method maintains 1× train / 1× inference with consistent SD reductions is compelling for real-world training.

**Weaknesses:**

1. Results are on CIFAR-10/100 with relatively small backbones, which naturally should have lower variance. It is unclear whether the same reductions hold on ImageNet, long-horizon training (e.g., ViTs), or self-supervised training (MAE, DINO) or language modeling (GPT-2)
2. There is little discussion when VML might over-regularize the variance and how to detect/avoid it (e.g., monitoring δ_t or λ_t statistics) would help practitioners.
3. There is no discussion regarding how the scaling model parameters might affect resulting gpu memory usage or training speed.

**Questions:**

1. Would the method behave less effectively if we use warmup learning rate or different learning rate schedules initially? It seems latch reference could heavily be impacted by the training recipe during early stage?
2. How would the method behave on larger dataset (like ImageNet) on other architectures (like ViT)? Does scaling on these dimensions still make the method same effective (1x training and seed variance reduction)?

---

### Note · Authors · 2026-01-15

I have read and agree with the venue's withdrawal policy on behalf of myself and my co-authors.